# Preparation of Polymer Composite Selective Permeable Membrane with Graphene Oxide and Application for Chemical Protective Clothing

Haolin Du [1], Zenghe Li [1,*], Heguo Li [2,*], Yue Zhao [2,*], Xiaopeng Li [1], Jing Liu [2] and Zuohui Ji [2]

[1] College of Chemistry, Beijing University of Chemical Technology, Beijing 100029, China; 2019210608@buct.edu.cn (H.D.); lxpbuct@163.com (X.L.)
[2] State Key Laboratory of NBC Protection for Civilian, Beijing 100191, China; liuj200721296@sina.com (J.L.); zuohui1210@163.com (Z.J.)
* Correspondence: lizh@mail.buct.edu.cn (Z.L.); liheguo1972@126.com (H.L.); SA11226532@mail.ustc.edu.cn (Y.Z.); Tel.: +86-010-64435714 (Z.L.)

**Abstract:** Chemical warfare agents (CWA) can poison people through the skin and cause injury, and the use of chemical protective clothing (CPC) is an important way to protect personnel from injury. CPC performance strongly depends on chemical protective materials, and satisfactory protective materials must meet various requirements, including protective performance, physiological comfort, mechanical performance, and cost effectiveness. Here, low-cost materials were used to prepare PVDF sodium sulfonate composite membranes with different contents of modified graphene oxide (GO-SSS). Their tensile properties, contact angle, permeability, and selectivity were tested and analyzed. The results show that when the addition ratio of GO-SSS to the bare membrane is 0.5%, the composite membrane has desirable permeation selectivity of water vapor/CWA simulant vapor and desirable mechanical properties. Hence, our sodium sulfonate composite membrane of PVDF with GO-SSS is an ideal material for potential applications in CPC.

**Keywords:** chemical protection; modified graphene oxide; nerve agent; water vapor penetration; microphase separation

## 1. Introduction

Chemical protective clothing (CPC) is a critical way to protect people from exposure to vapors of chemical warfare agents (CWAs). The best-known CWAs, nerve agents (represented by tabun (GA), sarin (GB), soman (GD), and VX) and blister agents (represented by sulfur mustard (HD)), are harmful to the skin. Nerve agents can cause acute poisoning and blister agents cause skin ulceration. Because of the danger of chemical agents, researchers use simulants to study them in the initial stage, and the typical simulants are 2-chloroethyl phenyl sulfide (CEPS) for blister agents and dimethyl methyl phosphonate (DMMP) for nerve agents.

Breathable and impermeable types are the main categories of CPC. The impermeable type of CPC generally uses barrier materials and has a reliable protection performance [1–3]. However, water vapor transmission is obstructed by the barrier materials simultaneously when blocking the CWAs, causing intense heat stress and even fainting [4]. In contrast, the breathable type of CPC, which is usually made of absorptive fiber, has much better physiological comfort [5,6]. However, the use of adsorbents brings problems such as bulkiness, limited adsorption capacity, poor environmental adaptability, and secondary contamination [7].

Polyelectrolyte membranes usually have a certain microphase separation structure. Under a swelling action, nano-scale water channels are formed in the membrane material, which is conducive to the rapid passage of water molecules, while the passage of poison

molecules is limited by the molecular size and molecular polarity. The diffusion coefficient in the membrane is very small, thereby realizing high-efficiency moisture permeability under the long protection time of the membrane material [8]. Ideally, the protective material should be selectively permeable, allowing water to pass through and blocking CWAs. Polyelectrolyte membranes (PEMs) represent an ideal material to solve this contradiction because of their unique microphase separation structure composed of hydrophobic and hydrophilic domains. Nanoscale hydrophilic channels can be formed through the self-assembly process and endow the PEM with selective permeability. The typical PEM materials widely reported so far for CPC applications are polystyrene sulfonic acid (PSSA), perfluorinated ionomers (Nafion), and sulfonated poly(styrene-isobutylene-styrene) tri-block copolymer (SIBS) [9–18]. However, these materials have their own weaknesses. The water/DMMP selectivity of sulfonated SIBS is relatively poor (0.7–35) [18], the cost of Nafion is too high for CPC applications (USD 1600 m$^{-2}$ for Nafion 117) [19], and the mechanical properties of PSSA are relatively poor [13].

Graphene is a typical two-dimensional material. It has excellent barrier properties and is considered to be the thinnest high-barrier material in the world. Oxygen-containing functional groups on the graphene oxide (GO) provide steric hindrance, which makes the interlayer spacing larger, and these oxygen-containing functional groups have a tendency of clustering and stacking, so the large gaps of unoxidized parts of the GO sheet are reserved. These gaps form a graphene capillary network inside the GO layers. The 2D graphene nanocapillary network allows the low friction slip of the monolayer water [20]. Due to the electroconductive characteristics of the polyelectrolyte membrane, the incorporation of unmodified GO into the membrane will cause serious phase separation. GO clusters are small particles and are not compatible in the polyelectrolyte membrane casting solution. Therefore, we adopt the method of polymerization grafting, using sodium p-styrene sulfonate (SSS) as the monomer to initiate polymerization on the surface of GO. After the vinyl monomer is initiated, macromolecular free radicals are immediately formed through chain growth. Some macromolecular free radicals are added to the double bonds of graphene oxide to generate graphene oxide-based brushes, and new free radicals are generated on the surface at the same time, so that the free radicals further grow, terminate, or transfer, and finally, the polyelectrolyte molecules brush improves the compatibility of GO [21].

In this work, we demonstrate the synthesis of a polymer polyvinylidene fluoride-grafted sodium polystyrene sulfonate (PVDF-g-SSS) using a facial one-pot process. In an attempt to improve the barrier performance of the polymer, sulfonic acid-modified GO was synthesized and introduced to the polymer membrane.

The obtained PVDF-g-SSS composite membrane was used for CPC material evaluation, and mechanical properties, contact angle, permeability, and selectivity differences were systematically studied. It was found that when the GO-SSS blending ratio was 0.5%, the GO composite membrane showed the optimum performance of CPC.

## 2. Materials and Methods

### 2.1. Materials

Dimethyl methyl phosphonate (DMMP), 2,2-azoisobutyronitrile (AIBN), divinylbenzene (DVB), tetramethyl hydroxide (TMAH), methanol, 2-chloroethyl phenyl sulfide (CEPS), sodium p-styrene sulfonate (SSS), graphene oxide (GO), N,N-dimethylformamide (DMF), and Polyvinylidene fluoride (PVDF) were all purchased from Sigma-Aldrich.

### 2.2. Membranes Preparation

GO (100 mg) and DMF (100 mL) were added and mixed in 200 mL Schlenk, and sonicated in 40 kHz sound waves for 1 h, and 50 mmol SSS monomer and 0.5 mmol AIBN were added to the mixture under N$_2$ flow protection and vigorous stirring. Then, the Schlenk was immersed in an oil bath at 65 °C. After 48 h of reaction, the product was collected and transferred to a centrifuge tube. The solution was centrifuged at 15,000 rpm for 0.5–1 h. We repeated the centrifugation (at least three times) until the upper layer was

colorless, and collected the black solid at the bottom. Finally, the product was freeze-dried for 48 h to obtain the final product, named GO-SSS.

PVDF-g-SSS membrane solution was synthesized through the one-pot process of Zhao et al. [8]. Briefly, we mixed 20 g Polyvinylidene fluoride (PVDF) with DMF, and then stirred the mixture at 50 °C until the mixture became uniform and transparent. We added 1.0 ml of TMAH methanol solution and 16 g SSS to the above mixture in sequence, followed by continuous stirring at 50 °C for an additional 1.5 h. Then, we added 1.6 g DVB crosslinker and 0.16 g AIBN initiator to the mixture and heated it to 80 °C for 8 h to obtain the PVDF-g-SSS polymer solution. Different masses of GO-SSS were added into the PVDF-g-SSS polymer solution to prepare composite membranes of different mass fractions (0%, 0.05%, 0.1%, 0.5%, 1%, 2%). Then, the blending polymer solutions were dried at 60 °C for 12 h. The obtained dry polymer membranes with different GO-SSS contents were named M1–M6.

### 2.3. Infrared Analysis

The Nicolet iS10 FT-IR Spectrometer (Thermo Fisher Scientific, Waltham, MA, USA) was applied to analyze the chemical properties of the materials. The conventional tablet pressing method was used for sample preparation.

### 2.4. Morphology Analysis

The JEOL JSM-7900F electron microscope (JEOL Ltd., Akishima, Tokyo, Japan) was applied to characterize the cross-section morphologies of the materials at a voltage of 20 kV. The cross-sections of the samples were created by liquid nitrogen freezing brittle fracture.

### 2.5. Water Uptake (WU), Ion Exchange Capacity (IEC), and Linear Swelling Ratio (LSR)

The IEC values of membrane samples were recorded using acid–base titration. First, 0.2 g of membrane was immersed in 1 M HCl for 24 h to make sure all counterions exchanged into $H^+$. Secondly, the dry membrane after washing with DI water was weighed with an analytical balance. Finally, the membrane was soaked in 1 M NaCl and equilibrated for 24 h. Using phenolphthalein as an indicator, 0.04 M NaOH aqueous solution was used to titrate the soaking solution, and the IEC value was calculated by Equation (1) based on the NaOH consumption in titration ($V_{NaOH} \times C_{NaOH}$) and the dry membrane weight in $H^+$ form ($W_{dry}$).

The dry membranes were cut into pieces of a certain size (10 mm × 30 mm). The samples were immersed in DI water for 24 h to reach a saturated water absorption state. WU and LSR were calculated by Equations (2) and (3) based on the weight and length differences between the samples under wet and dry states, respectively.

$$IEC = \frac{V(NaOH) \times C(NaOH)}{W_{dry}} \tag{1}$$

$$WU = \frac{W_{wet} - W_{dry}}{W_{dry}} \times 100\% \tag{2}$$

$$LSR = \frac{L_{wet} - L_{dry}}{L_{dry}} \times 100\% \tag{3}$$

where $W_{dry}$ and $W_{wet}$ represent the weights of the samples in the dry and wet states, respectively, and $L_{dry}$ and $L_{wet}$ represent the lengths of the samples in the dry and wet states, respectively.

### 2.6. Wettability

The wettability of the membranes was evaluated by using a contact angle meter fabricated by Solon Tech Co., Ltd (Shenzhen, China). The measurement of the static contact

angle was to drop DI water onto the sample surface at room temperature [22]. The accuracy of the results was ensured by three parallel measurements.

### 2.7. Mechanical Strength Test

The membranes were cut into pieces with size of 5 mm × 30 mm, and the mechanical properties were studied by tensile testing using an Instron Universal Testing Machine (Instron Co., Canton, MA, USA). The accuracy of the results was ensured by three parallel measurements.

### 2.8. Permeations and Selectivity Tests

The vapor permeation measurement of the membrane adopted the American Society for Testing and Materials (ASTM) E96-95 method with a few changes. Before the test, the membrane was dried at 60 °C for 24 h, and a digital caliper was used to measure the thickness of the membrane. Then, the test cell was filled with a certain quality of penetrant (water, DMMP, or CEPS), and the membrane to be tested was placed over the top of the open-top testing cell and fixed with a fixing device to form a permeation cell, as shown in Figure 1. The obtained permeation cell was put into a convection environment box with constant temperature of 35 °C and humidity of 10% RH and weighed regularly. The weight loss (*W*) was calculated from the slope of the weight–time curve after the weight loss became constant. The vapor transmission rate (VTR, g m$^{-2}$ days$^{-1}$) was calculated by Equation (4) and the vapor permeability (VP, mol m$^{-1}$ s$^{-1}$) was calculated by Equation (5). The selectivity was defined as the ratio of vapor permeability of water to DMMP or CEPS, and the value of selectivity was calculated by Equation (6) [14]. The accuracy of the results was ensured by three parallel measurements.

$$\text{VTR} = W/(t \times A) \tag{4}$$

$$\text{VP} = \text{VTR} \times L \tag{5}$$

$$\text{S} = \text{VP}_{water}/\text{VP}_{simulants} \tag{6}$$

where *A* represents the tested area of the sample, *t* represents the test time, and *L* is the membrane thickness.

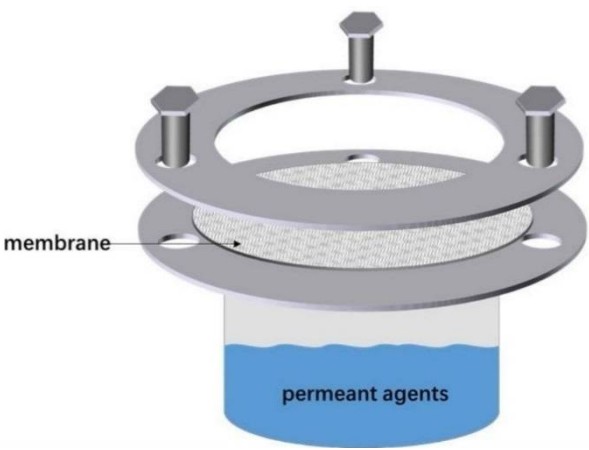

**Figure 1.** Schematic drawing of the permeation cell.

### 3. Results

#### 3.1. Synthesis and Characterizations of GO-SSS and PVDF-g-SSS Composite Membrane

As shown in Figure 2a, small clusters existed in the casting solution, and GO was not compatible when 0.5% unmodified GO was directly mixed with the PVDF-g-SSS casting solution. In Figure 2b, it can be seen that the casting solution doped with GO-SSS had

a darker color, was more uniform, and had no cluster existence compared to the casting solution doped with the same amount of GO.

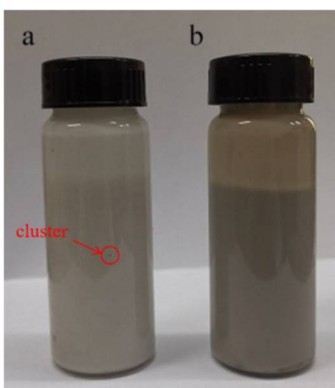

**Figure 2.** (**a**) GO mixed PVDF-g-SSS membrane solution, (**b**) GO-SSS mixed with PVDF-g-SSS membrane solution at the same concentration of 0.5%.

Figure 3 shows the photographs of M1–M6 membranes, which are quite uniform. It was noticed that the higher the proportion of GO-SSS incorporated, the darker the color of the membrane material.

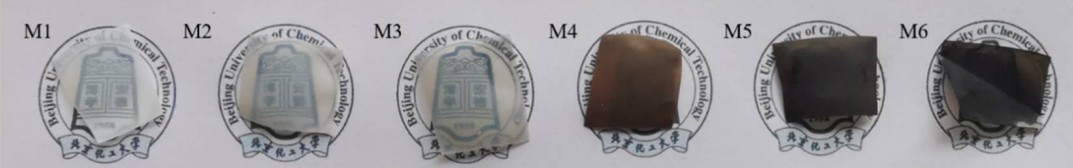

**Figure 3.** The morphology of M1–M6 membranes.

Figure 4 depicts the FT-IR spectra of GO and GO-SSS powders after freeze-drying. Compared with bare GO powder (a), the freeze-dried GO-SSS powder (b) shows absorption bands at 1300 cm$^{-1}$ to 1550 cm$^{-1}$, representing the benzene ring's skeleton stretching vibration. The absorption peak shown at 2800 cm$^{-1}$ to 3000 cm$^{-1}$ represents the stretching vibration peak of the methylene group on the benzene ring. In the FTIR spectrum of GO-SSS powder, the absorption peaks at 1010 cm$^{-1}$ and 1150 cm$^{-1}$ represent the symmetric stretching vibration peaks of the sulfonate [23]. This indicates that the surface of GO has been successfully grafted with sodium p-styrene sulfonate.

Figure 5a shows that GO shows a strong diffraction peak at 2θ = 11.34°, indicating that the interlayer spacing based on Bragg's law (2dsinθ = n × λ) is about 0.78 n. In comparison, no obvious sharp diffraction peak was observed in the case of GO-SSS (Figure 5b), indicating that the polyelectrolyte molecular brush destroyed the periodic structure of GO and effectively inhibited the aggregation of the graphene sheets. The original GO's sharp diffraction peak at 2θ = 11.34° shifted to 2θ < 5.0° after GO modification, and the corresponding interlayer spacing was greatly increased from 0.78 nm to above 1.75 nm. These results indicate that the polyelectrolyte molecular brush was successfully grafted onto the surface of the GO sheet.

The SEM image in Figure 6 shows the topography of GO and GO-SSS. Figure 6a clearly shows the lamella structure of GO, while Figure 6b displays the hairy two-dimensional molecular brush flaky structure of GO-SSS.

The SEM image in Figure 7 shows the section surface morphology of the original PVDF membrane and the composite membrane. Clearly, the surface of the original membrane is dense and non-porous, while the composite membrane has many wrinkles on the surface due to the presence of GO-SSS sheets.

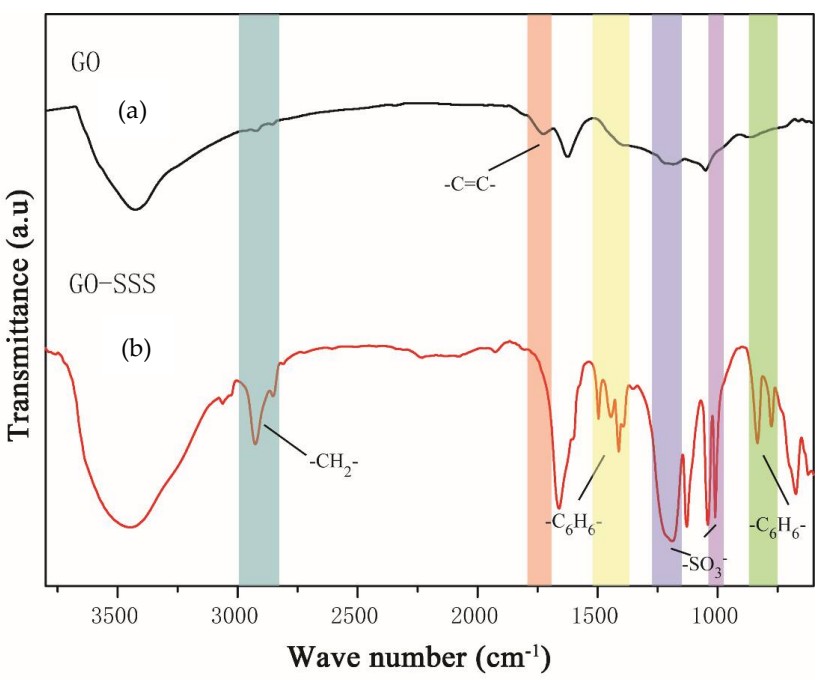

**Figure 4.** FT-IR spectra of (**a**) GO, (**b**) modified GO.

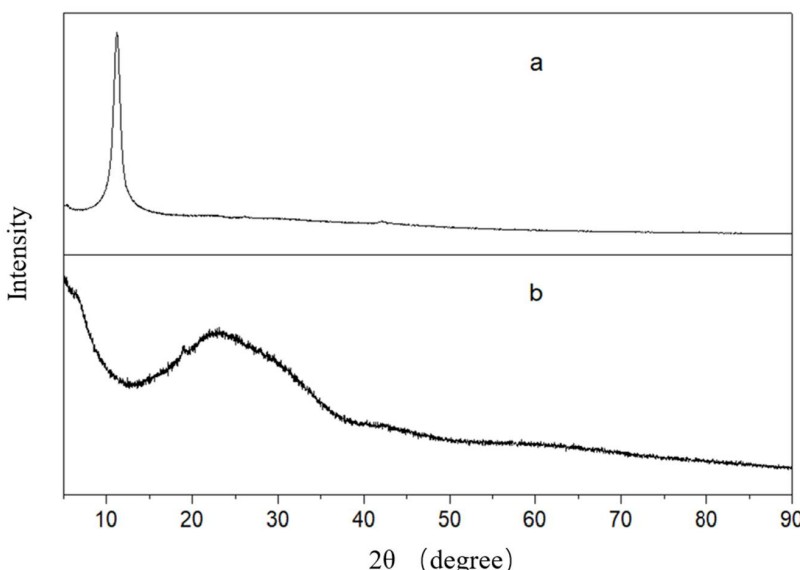

**Figure 5.** XRD pattern of (**a**) GO, (**b**) GO-SSS.

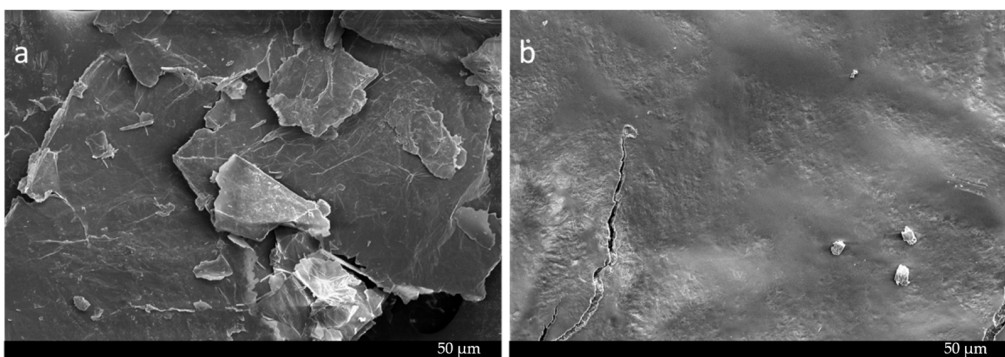

**Figure 6.** SEM image of (**a**) GO surface, (**b**) GO-SSS surface.

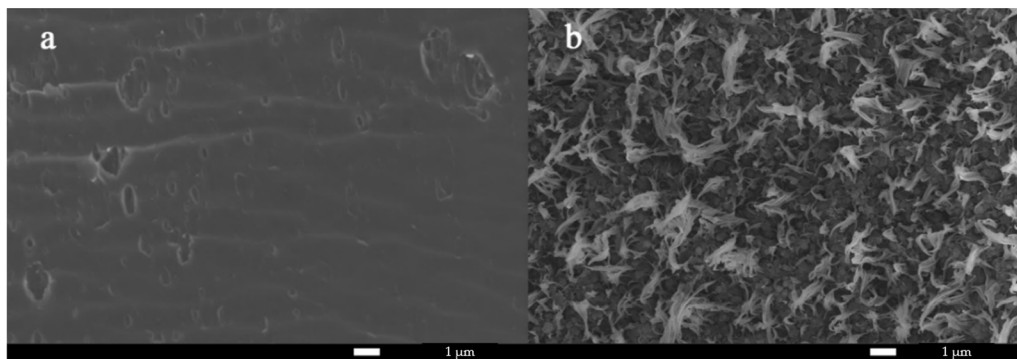

**Figure 7.** SEM image of cross section of (**a**) bare PVDF-SSS membrane, (**b**) composite membrane.

### 3.2. IEC, WU, and LSR

IEC is a key attribute of PEM and can affect WU, LSR, and water/DMMP selectivity. As shown in Table 1, the IEC value decreased slightly from M1 to M6, and the WU and LSR decrease with the decreases of IEC.

**Table 1.** WU, ICE, LSR, and WCA.

| Sample | IEC (mmol/g) | WU (%) | LSR (%) | CA of Water (°) |
|--------|--------------|--------|---------|------------------|
| M1 | 1.97 | 53.6 | 16. 8 | 40.1 ± 2.2 |
| M2 | 1.95 | 53.2 | 16.3 | 44.9 ± 2.6 |
| M3 | 1.90 | 53.1 | 16.1 | 50.2 ± 2.3 |
| M4 | 1.88 | 52.8 | 15.8 | 55.0 ± 2.4 |
| M5 | 1.83 | 52.4 | 15.6 | 58.3 ± 2.1 |
| M6 | 1.80 | 51.6 | 15.2 | 60.6 ± 1.8 |

### 3.3. Water Contact Angle (WCA)

Figure 8 and Table 2 show the WCA of the synthetic membrane. As the membrane material has lower surface hydrophilicity under low IEC, the water contact angle increases from M1 to M6, and the hydrophilicity gradually decreases, which is consistent with the trend of WU. These results prove that the incorporation of modified GO reduces the hydrophilicity of the membrane.

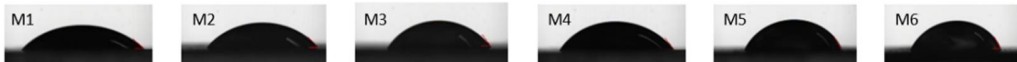

**Figure 8.** WCA of M1–M6 membranes.

**Table 2.** Thickness and mechanical properties of PVDF-g-SSS membranes.

| Sample | Thickness (μm) | Strength (MPa) | Elongation at Break (%) |
|--------|----------------|----------------|--------------------------|
| M1 | 120 | 17.5 ± 2.3 | 235.3 ± 18.8 |
| M2 | 117 | 13.2 ± 2.2 | 113.7 ± 14.2 |
| M3 | 113 | 15.3 ± 2.3 | 102.4 ± 19.5 |
| M4 | 109 | 13.7 ± 1.5 | 60.6 ± 15.4 |
| M5 | 106 | 14.1 ± 1.3 | 46.4 ± 8.4 |
| M6 | 103 | 12.9 ± 1.6 | 26.8 ± 5.3 |

### 3.4. Mechanical Properties

For CPC, the mechanical strength of the membrane should be able to withstand the damage caused by the strenuous action and washing. Figure 9 is the stress–strain curve of M1–M6 membranes. The elongation at break decreases with the increase in the content of GO-SSS, that is, as the amount of added GO-SSS increases, the flexibility of

the film decreases. The strength of the membrane has no obvious trend. The overall strength of the membrane doped with GO-SSS is not as strong as the original PVDF-g-SSS membrane. Among the GO-SSS-doped membranes, M3 has relatively higher tensile force and elongation at break and better comprehensive mechanical properties. As seen from Figure 9, the slope of the linear part (elastic strain) of the tension curve increases with the increase in GO-SSS dosage, indicating that the Young's modulus of the membrane increases with higher content of GO-SSS. With the increase, the rigidity of the membrane becomes stronger, that is, it is less prone to suffer deformation, which is detrimental for the application of protective clothing. The tensile strength and elongation at break of the film are summarized in Table 2. In comparison to M1 (the original membrane without GO-SSS), the overall membrane material becomes more rigid and easier to break with the increase in GO-SSS dosage.

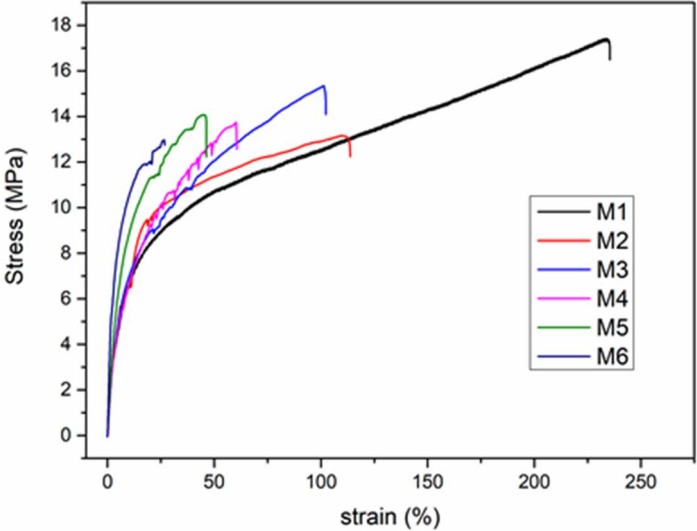

**Figure 9.** Typical stress–strain curves of M1–M6 membranes.

### 3.5. Permeations and Selectivity of Membranes

In the application of CPC, the transmission rate of water vapor should be higher than 2000 g m$^{-2}$·day$^{-1}$ [3]. As shown in Table 3, all the synthetic membranes M1–M6 possessed a WVTR value greater than 2000 g m$^{-2}$·day$^{-1}$, meeting the CPC requirement. The sodium sulfonate groups within the membrane has satisfactory affinity for water molecules, so it can promote water diffusion and improve the moisture permeability of the membrane by increasing the hydrophilicity.

**Table 3.** VTR of water; VP of water, DMMP, and CEPS; selectivity between water and DMMP; and selectivity between water and CEPS of M1–M6 membranes.

| Sample | VTR of Water (g m$^{-2}$ 24 h$^{-1}$) | VP of Water × 10$^{-9}$(mol s$^{-1}$ m$^{-1}$) | VP of DMMP × 10$^{-9}$ (mol s$^{-1}$ m$^{-1}$) | VP of CEPS × 10$^{-9}$ (mol s$^{-1}$ m$^{-1}$) | Selectivity (Water/DMMP) | Selectivity (Water/CEPS) |
|---|---|---|---|---|---|---|
| M1 | 3307 ± 75 | 255.23 ± 5.76 | 2.15 ± 0.18 | 0.13 ± 0.01 | 118.71 ± 7.26 | 1963.31 ± 106.72 |
| M2 | 3235 ± 64 | 243.36 ± 4.82 | 2.03 ± 0.12 | 0.12 ± 0.02 | 119.88 ± 4.71 | 2028.00 ± 297.83 |
| M3 | 2902 ± 55 | 210.89 ± 4.03 | 1.73 ± 0.06 | 0.10 ± 0.01 | 121.90 ± 1.89 | 2108.90 ± 170.59 |
| M4 | 2622 ± 52 | 183.78 ± 3.63 | 1.46 ± 0.08 | 0.08 ± 0.01 | 125.88 ± 4.41 | 2297.25 ± 241.78 |
| M5 | 2593 ± 51 | 176.74 ± 3.51 | 1.67 ± 0.11 | 0.12 ± 0.02 | 105.83 ± 4.87 | 1472.83 ± 216.22 |
| M6 | 2652 ± 56 | 175.66 ± 3.74 | 1.89 ± 0.13 | 0.14 ± 0.02 | 92.94 ± 4.41 | 1254.71 ± 152.53 |

The vapor permeation (VP) of DMMP is shown in Figure 10 and Table 3. When the addition ratio of GO-SSS increased from 0% to 2%, the DMMP vapor permeation decreased first and then increased. The selectivity between water vapor and DMMP is displayed in Figure 11. When the addition ratio of GO-SSS is from 0% to 0.5%, the selectivity gradually increases from 118 to 125. The composite membrane shows higher selectivity than the

PSSA membrane (80) and the sulfonated poly(styrene-isobutylene-styrene) membrane (35) reported elsewhere [13,18], suggesting the role of GO-SSS in the improvement of membrane selectivity of water/DMMP.

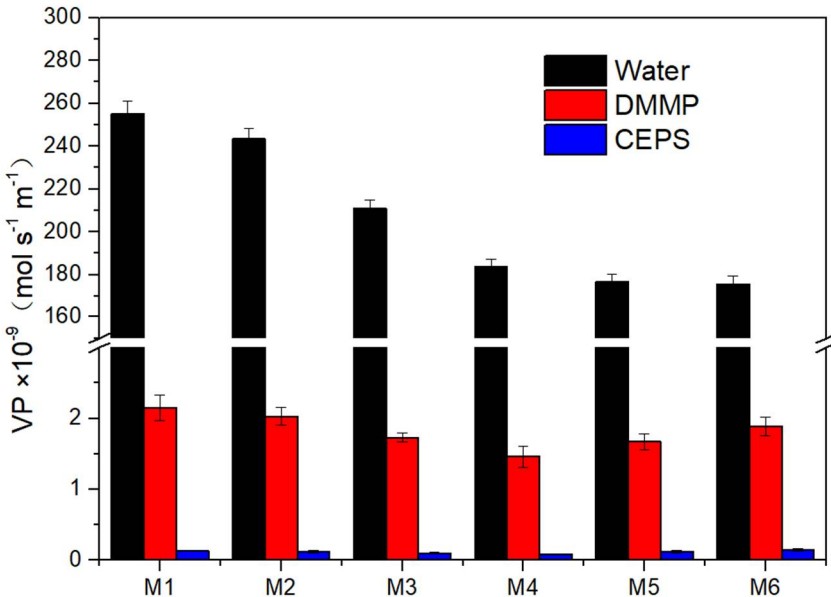

**Figure 10.** Vapor permeation of water, DMMP, and CEPS through M1–M6.

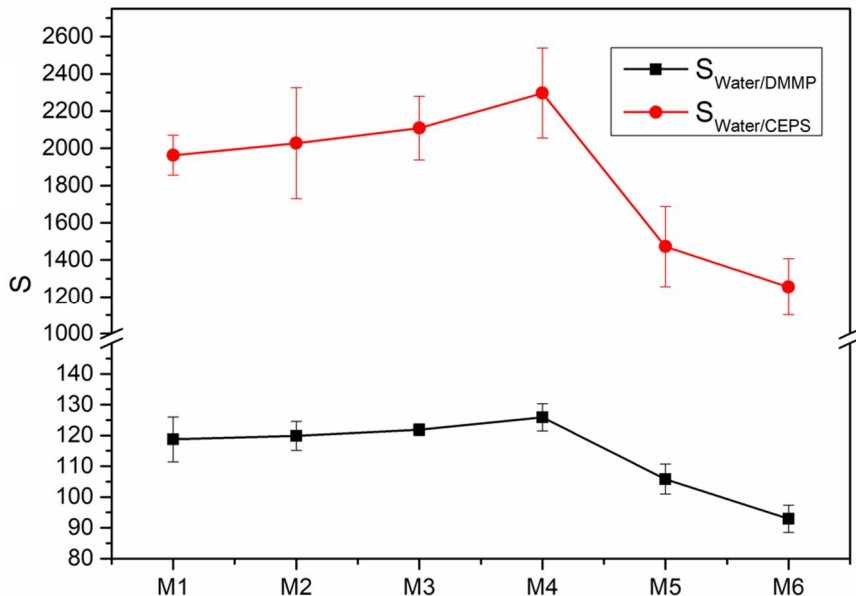

**Figure 11.** Selectivity of water/DMMP and water/CEPS for M1–M6.

The vapor permeability (VP) of CEPS is shown in Figure 10 and Table 3. CEPS vapor permeability first increased and then decreased with GO-SSS dosage. When the addition ratio of GO-SSS is 0.5%, the selectivity gradually increases from 1967 to 2293. This shows that the addition of GO-SSS can significantly improve the selectivity of water/CEPS, which is much higher than the PSSA membrane (1500) and Nafion membrane (400) [13].

## 4. Discussion

Nafion and PSSA have been regarded as the most advanced membrane materials for CPC so far. Unfortunately, Nafion is not cost-effective for large-scale applications, and PSSA shows undesirable mechanical properties. Hence, we prepared a series of PVDF-g-SSS

membranes with different contents of GO-SSS (M1–M6) using a one-pot process. FT-IR, SEM, and XRD clearly proved the success of SSS grafting onto GO.

The SEM image of GO and GO-SSS clearly shows that through in situ free radical polymerization, the polymer chain has been successfully grafted onto the GO sheet. The morphology of the membrane is uniform, dense, and non-porous. Generally, there are two mechanisms to control the penetration of substances in separation membrane materials. One is to screen substances of different sizes through their own pore structure, such as the microfiltration membrane and the nanofiltration membrane. The other is to screen substances with different physical and chemical properties through their own dissolution and diffusion properties, such as the water–oil separation membrane [24]. The SEM image of the composite membrane indicates that the selectivity of the membrane is not due to the pore size screening mechanism, but due to the dissolution–diffusion mechanism.

The IEC measurement shows that the conversion rate of the SSS monomer is very high (>70%). It was found that WU and LSR decrease nonlinearly with the decrease in IEC. Contact angle measurement showed that doping with different contents of GO-SSS can significantly reduce the hydrophilicity of the membrane.

The WVTR test shows that with the increase in the GO-SSS ratio, the WVTR of the composite membrane decreases, but the WVTR of the M1–M6 composite membranes exceeds $2000 \, \text{g} \, \text{m}^{-2} \, \text{days}^{-1}$, which can meet the requirements of CPC applications. The steam permeation test results show that when the GO-SSS addition ratio is 0.5%, the composite membrane can obtain a water/DMMP selectivity greater than 120 and a water/CEPS selectivity greater than 2200. The excellent selectivity of high-IEC membranes is due to the formation of interconnected SSS phases that allow water transmission while preventing the transmission of DMMP. Vapor penetration occurs through two steps: vapor adsorption and vapor diffusion [25]. The SSS group can form hydrated regions, namely water channels, which can improve the absorption of water. Simultaneously, hydrophobic CWA simulants are not favored by the water channels, leading to low absorption. On the other hand, water diffuses mainly through the hydrophilic phase, while DMMP diffuses mainly through the hydrophobic phase [9]. The greater the proportion of GO-SSS, the better the barrier properties of the composite film. However, as the doping amount of GO-SSS increases, the VP of water decreases, which causes the selectivity of the membrane to increase first and then decrease, so the optimal doping ratio of GO-SSS is 0.5%.

## 5. Conclusions

This work reveals that PVDF-g-SSS doped with GO-SSS has the potential for CPC applications, and the GO-SSS dosage in the membrane determines the overall performance of the composite membrane. Although a high GO-SSS blending ratio is beneficial for the improvement of the barrier properties of the composite membrane, the high blending amount inevitably reduces the water vapor permeation properties of the composite membrane. On the contrary, the composite membrane with a low GO-SSS doping ratio has a higher water vapor permeation rate. Therefore, controlling the amount of the GO-SSS addition can effectively control the properties of membrane materials to achieve a balanced overall performance in CPC applications. For the composite membrane in this work, the optimal addition amount of GO-SSS is determined as 0.5%, which can provide 2622 WVTR and 125 water/DMMP selectivity, and 2297 water/CEPS selectivity.

**Author Contributions:** Conceptualization, Y.Z., H.L. and Z.L.; methodology, Y.Z., H.D. and X.L.; formal analysis, Y.Z. and H.D.; original draft preparation, Y.Z.; review and editing, H.L., J.L. and Z.J. All authors have read and agreed to the published version of the manuscript.

**Funding:** This research was funded by the National Defense Basic Scientific Enhancement program, grant number 2019-JCJQ-JJ-160.

**Institutional Review Board Statement:** Not applicable.

**Informed Consent Statement:** Not applicable.

**Data Availability Statement:** Not applicable.

**Acknowledgments:** The testing part of this work was supported by the Beijing Center for Physical and Chemical Analysis.

**Conflicts of Interest:** The authors declare no conflict of interest.

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
