# Peer review of "Preparation of Polymer Composite Selective Permeable Membrane with Graphene Oxide and Application for Chemical Protective Clothing"

_processes, doi:10.3390/pr10030471_

Round 1

Reviewer 1 Report

The manuscript entitled "Preparation of Graphene Composite Selective Permeable  Membrane and Application of Chemical Protective Clothing" deals with a fabrication of PVDF sodium sulfonate composite membranes with graphene oxide composite. The authors demonstrated that the addition ratio of GO-SSS to the bare membrane enhanced its the mechanical properties in one hand. And in other hand, this addition gave to the membrane composite a desirable selectivity of water vapor/ CWA simulant vapor permeation. Put in all together, the authors showed the capability of graphene oxide to enhence the properties of PVDF sodium sulfonate membrane, resulting in the fact that the sodium sulfonate composite membrane of PVDF with GO-SSS can be used potentially as a Chemical protective clothing material.  Therefore, I can recommend the manuscript for publication after a minor revision is done addressing the points below:

1-The author should make a more accurate description of the topic for the title.

I propose some thing like that:

Preparation of polymer Composite Selective Permeable Membrane with Graphene oxide and Application for Chemical Protective Clothing.

I propose the above title because in their study, the polymer PVDF is the matrix of the composite and the graphene oxide is used as the reinforcement. And also it is better to say graphene oxide instead of graphene because they have not the same properties.

2- Figure 5: Can the authors add the Y axis in this Figure? The authors should also comment in figure 5b in text.

Reviewer 2 Report

In general the manuscript is interesting, the results are in accordance with the methodology used, they are clear and well analyzed. The present manuscript consists of obtaining composite materials, in which modified graphene oxide is used as an alternative for the synthesis of materials with better mechanical properties for their use in safety clothing against hazardous substances. The following recommendations are suggested:

  1. It is recommended to place the meaning of some acronyms such as PVDF.
  2. It is suggested not to start sentences with numbers, e.g.: line 89
  3. To explain in more detail some aspects of the methodology used in terms of sample preparation and FT-IR and morphology analysis. 
  4. In standardized methodologies such as contact angle, provide the necessary bibliography to establish how it was developed.
  5. In the results item, parts of the methodology are repeated, see lines 161-162, it is recommended to eliminate them.
  6. Maintain uniformity in the name of the prepared materials, see line 189 (modified GO or GO-SSS).
  7. Replace Fig. 4b with Fig. 5b. see line 194
  8. In the Figure 5, Y-axis have not title 
  9. It is recommended to write the sentence on lines 205-207 later in the manuscript when this statement is evident with reference to the results.
  10. It is recommended to extend the idea of lines 288-289.
